# Savory Signaling: T1R Umami Receptor Modulates Endoplasmic Reticulum Calcium Store Content and Release Dynamics in Airway Epithelial Cells

**DOI:** 10.3390/nu15030493

**Published:** 2023-01-18

**Authors:** Derek B. McMahon, Jennifer F. Jolivert, Li Eon Kuek, Nithin D. Adappa, James N. Palmer, Robert J. Lee

**Affiliations:** 1Department of Otorhinolaryngology, Perelman School of Medicine, University of Pennsylvania, Philadelphia, PA 19104, USA; 2Department of Physiology, Perelman School of Medicine, University of Pennsylvania, Philadelphia, PA 19104, USA

**Keywords:** T1R1, T1R3, umami, amino acids, apoptosis, airway

## Abstract

T1Rs are expressed in solitary chemosensory cells of the upper airway where they detect apical glucose levels and repress bitter taste receptor Ca^2+^ signaling pathways. Microbial growth leads to a decrease in apical glucose levels. T1Rs detect this change and liberate bitter taste receptor signaling, initiating an innate immune response to both kill and expel pathogens through releasing antimicrobial peptides and increasing nitric oxide production and ciliary beat frequency. However, chronic inflammation due to disease, smoking, or viral infections causes a remodeling of the epithelial airway. The resulting squamous metaplasia causes a loss of multi-ciliated cells and solitary chemosensory cells, replaced by basal epithelial cells. To understand how T1R function is altered during disease, we used basal epithelial cells as a model to study the function of T1R3 on Ca^2+^ signaling dynamics. We found that both T1R1 and T1R3 detect amino acids and signal via cAMP, increasing the responsiveness of the cells to Ca^2+^ signaling stimuli. Either knocking down T1R1/3 or treating wild-type cells with MEM amino acids caused a reduction in ER Ca^2+^ content through a non-cAMP signaled pathway. Treatment with amino acids led to a reduction in downstream denatonium-induced Ca^2+^-signaled caspase activity. Thus, amino acids may be used to reduce unwanted apoptosis signaling in treatments containing bitter compounds.

## 1. Introduction

Upper airway epithelial cells initiate an innate immune response through taste receptor signaling pathways in retaliation against inhaled pathogens. Bitter taste receptors (T2Rs) localized to the motile cilia of multi-ciliated airway epithelial cells [1] detect “bitter” bacterial quorum sensing molecules such as quinolones and *N*-Acyl homoserine lactones [2,3]. While mucus provides a physical web to ensnare pathogens, T2Rs trigger a calcium (Ca^2+^) signaling cascade leading to an increase in nitric oxide (NO) production while also increasing ciliary beat frequency to hasten mucociliary clearance, thus providing a means to both kill and remove ensnared pathogens [4,5].

Pathogens feed on apical glucose in the airway surface liquid. Solitary chemosensory cells (SCCs) are a rare epithelial cell type that contain both T2Rs and sweet taste receptors (T1Rs) [3,6,7,8]. In type 2 taste cells on the tongue, T1R1 and T1R3 dimerize to form a receptor that interacts with L-amino acids and small peptides to transduce signaling pathways leading to the sensation of umami [9] while T1R2 and T1R3 dimerize to form a taste receptor interacting with sweet-tasting compounds [10,11]. On the tongue, T1R’s signal through a heterotrimeric G protein complex consisting of a Gα-gustducin, which activates phosphodiesterase, and Gβ_3_, and Gγ_13_, which signal for intracellular Ca^2+^ elevations [12,13,14,15,16]. The release of the Gβγ dimer signals phospholipase Cβ2 and downstream activation of the TRPM5 channel [16,17]. This pathway ultimately leads to ATP release, stimulation of afferent gustatory neurons, and the sensation of taste.

In airway SCC’s, T1R2 and 3 are expressed [18] and hypothesized to repress T2R calcium (Ca^2+^) signaling through detection of apical glucose [19]. Once pathogens grow and reduce apical glucose concentrations, T1R2/3 no longer represses T2R signaling within the SCC [19]. T2Rs then cause a Ca^2+^-dependent release of anti-microbial peptides from neighboring multi-ciliated cells into the apical lumen of the airway to combat infections. Thus, in a typical, heathy epithelial layer, T1Rs and T2Rs function as sentinels of the upper airway.

However, chronic inflammation, caused by diseases such as chronic rhinosinusitis (CRS) [20], COPD [21], smoking [22], or severe asthma [23], causes a loss of cilia, reviewed in [24]. CRS is often characterized by squamous metaplasia, causing a loss of multi-ciliated cells, replacing the epithelial layer with squamous and basal epithelial cells instead [25,26]. Basal airway epithelial cells do retain some ability to produce nitric oxide in response to bitter compounds [27]. Because NO is an important antimicrobial that can damage bacterial DNA [28,29] as well as an anti-viral, -fungal, and -parasitic agent [30], therapeutics enhancing host NO production may be desirable. We hypothesize that topical therapies to boost NO production may be of use in chronic rhinosinusitis, but these therapies must be optimized to function even in diseased epithelia that have lost cilia.

While bitter compounds increase NO production, we have also shown that T2R activation induces apoptosis in squamous and/or basal airway cells [27]. Moreover, many clinical drugs (e.g., antihistamines) are known to be bitter, and thus might inadvertently activate apoptosis in some settings (e.g., as a nasal spray) when the epithelium is remodeled by inflammation. We sought to identify targets to protect airway cells from apoptotic responses activated by T2R agonists so they can be better used as therapeutics. Here, we show that both T1R1 and T1R3 function in airway squamous and basal cells to detect amino acids. The addition of amino acids decreases denatonium-signaled caspase activity, likely through the reduction of ER Ca^2+^ stores. Thus, umami receptor T1R1/3 may play a pivotal role in reducing apoptosis activated by bitter drugs in infected airway cells.

## 2. Materials and Methods

For a detailed list of reagents used in this work, please refer to Appendix A.

### 2.1. Western Blot

For endogenous T1R1 and T1R3 immunoblot visualization, cell pellets were resuspended in RIPA Buffer (50 mM Tris pH 7.5, 150 mM NaCl, 1% IGEPAL CA-630, 1% deoxycholate, 1 mM DTT, DNase, and Roche cOmplete Protease Inhibitor Cocktail) and lysed via sonicating water bath. Protein concentrations of post 800× *g* lysates were estimated by Bradford DC Assay (BioRad Laboratories, Hercules, CA, USA) then 60 µg protein/lane was loaded into a 4-12% Bis-Tris gel. After electrophoresis, the resulting gel was transferred to nitrocellulose and the membrane was blocked in 5% milk in Tris-Tween (50 mM Tris, 150 mM NaCl, and 0.025% Tween-20) for at least 1 h. The primary antibody was diluted 1:1000 in Tris-Tween containing 5% bovine serum albumin and incubated for at least 2 h. The secondary antibody goat anti-rabbit IgG-horseradish peroxidase was diluted 1:1000 in Tris-Tween containing 5% milk. After extensive washing, blots were incubated in Clarity ECL (BioRad Laboratories, Hercules, CA, USA) as per manufacturers’ protocol, and resulting immunoblots were visualized using a ChemiDoc MP Imaging System (BioRad Laboratories, Hercules, CA, USA). Final images were processed using Image Lab Software (BioRad Laboratories, Hercules, CA, USA).

### 2.2. Live Cell Imaging

For intracellular Ca^2+^ measurements, cells were loaded with either 5 µM of Fluo-8 AM or Fura-2 AM for 1 h in HEPES-buffered Hank’s Balanced Salt Solution in the dark at room temperature. For intracellular or nuclear cAMP, biosensors were transfected 48 h prior to measurements. Fura-2 AM images were taken using Fura-2 filters (79002-ET Chroma, Rockingham, VT, USA) while all other images were taken utilizing FITC or TRITC filter sets (49002-ET or 49004-ET Chroma) on an Olympus IX-83 microscope (20× 0.75 NA objective) with fluorescence xenon lamp and X-Cite 120 LED boost lamp (Excelitas Technology, Waltham, MA, USA), excitation and emission filter wheels (Sutter Instruments, Novato, CA, USA), Orca Flash 4.0 sCMOS camera (Hamamatsu, Tokyo, Japan) and MetaFluor software (Molecular Devices, Sunnyvale, CA, USA).

### 2.3. Human Primary Cell Culture

Primary human sinonasal cells were isolated from residual surgical material acquired in accordance with the University of Pennsylvania guidelines for use of residual clinical material along with the U.S. Department of Health and Human Services code of federal regulation (Title 45 CFR 46.116) and the Declaration of Helsinki. Informed consent from each patient was obtained in addition to institutional review board approval (#800614). Patients were >18 years old and either receiving surgery for sinonasal disease or other procedures.

Primary nasal epithelial cells were obtained via enzymatic digestion of sinonasal tissue then cultured as previous described [4]. Briefly, residual tissue was incubated in medium containing 1.4 mg/mL protease and 0.1 mg/mL DNase for 1 h at 37 °C. Proteases were neutralized with medium containing 10% FBS and resulting suspension of cells were incubated in a flask at 37 °C 5% CO_2_ for 2 h in PneumaCult-Ex Plus medium (Cat. 05040, StemCell Technologies) with 100 U/mL penicillin and 100 µg/mL streptomycin to remove non-epithelial cells. After 2 h, unattached cells were then plated into a tissue culture dish and propagated in PneumaCult-Ex Plus medium. Basal cell identity was confirmed by positive staining for basal cell markers CD49f and p63 (Appendix A). For differentiation, cells were grown in air-liquid interface (ALI) cultures for at least 7 days exposure to air prior to use in experiments using PneumaCult-ALI medium (StemCell Technologies Cat #05001). De-identified primary normal human bronchial epithelial cells from three individual donors were purchased from Lonza (Walkerville, MD, USA; Cat #CC-2541) and cultured the same as submerged primary nasal cells.

### 2.4. Knockdown of T1R Isoforms

Subconfluent Beas-2B cells were grown in F-12K Media with 10% FBS and 1% penicillin/streptomycin (Gibco) and transfected with either 10 nM of non-targeting, TAS1R1.6, or TAS1R3.2 RNAi duplex (Integrated DNA Technologies) for 48 h using lipofectamine 3000 (ThermoFisher Scientific, Waltham, MA USA) as per manufacturer’s protocol. Knockdown was validated using qPCR as described below.

### 2.5. Quantitative PCR (qPCR)

Cultured cells were resuspended in TRIzol (ThermoFisher Scientific, Waltham, MA USA) and either stored at −70 °C or immediately used. RNA was purified via Direct-zol RNA kit (Zymo Research, Irvine, CA, USA) subjected to RT-PCR using High-Capacity cDNA Reverse Transcription Kit (ThermoFisher Scientific). Resulting cDNA was then utilized for qPCR using Taqman Q-PCR probes in QuantStudio 5 Real-Time PCR System (ThermoFisher Scientific). Data was analyzed using Microsoft Excel and graphed in GraphPad PRISM v8.

### 2.6. Data Analysis and Statistics

For comparisons of only two data sets *t*-tests was used. ANOVA was utilized for more than two comparisons where Tukey–Kramer’s post-test was used for comparisons of all samples, Dunnett’s post-test was used to compare multiple samples to one control, and Bonferonni’s post-test was used for selective pair comparisons. In all figures, (*) *p* < 0.05, (**) *p* < 0.01, (***) *p* < 0.001, (****) *p* < 0.0001, “n.s.” (no statistical significance). All data represent the mean ± SEM of a minimum of 3 experiments.

## 3. Results

### 3.1. T1R3 Is Expressed in Airway Cells and Signals through cAMP

While previous work established a fundamental understanding of T1R activity in SCCs of fully differentiated airway epithelial cell cultures [6,7,8,18,19], their expression or activity in disease states remains unexplored. We used both the non-tumorigenic lung epithelial cell line Beas-2B’s and proliferating primary non-ciliated basal airway epithelial cells as a disease model along with differentiated primary airway cells, cultured in air-liquid interface as a healthy model. Quantified by qPCR, low levels of transcript for *TAS1R1*, *TAS1R2*, and *TAS1R3*, the genes encoding T1R1, T1R2, and T1R3, respectively, were expressed in both the Beas-2B and primary cells in both basal and differentiated cultures (Figure 1a). There was a comparable level of transcription between the Beas-2B cells and primary basal cells, suggesting that Beas-2B’s are a comparable model to primary basal epithelial cells in terms of T1R signaling. Interestingly, there was a 10-fold increase in *TAS1R1* transcription in differentiated cultures relative to basal primary cells (Figure 1a). While transcription levels of TAS1Rs were low compared with housekeeping genes, this is typical for receptors and was within the range of expression of TAS2R bitter receptor and toll-like receptor gene expression (Appendix A). Additionally, in both Beas-2B’s and primary basal cells, we have detected T1R1 and T1R3 protein expression via Western blot (Figure 1b).

Both T1R1 and T1R3 were previously reported to detect a broad range of amino acids and signal through intracellular Ca^2+^ to activate ERK1/2 and mTORC1 in MIN6 pancreatic β cells [31,32]. To evaluate T1R1/3 activity in airway epithelial cells, we first looked for intracellular Ca^2+^ release via addition of amino acids to see if similar pathways are present in our model. Utilizing the intracellular Ca^2+^ dye Fluo-8 and treating Beas-2B cells with a mixture of 1× MEM amino acids (MEM AA) and 1× non-essential amino acids (NEAAs) (Appendix A), we did not observe any changes in intracellular Ca^2+^ signaling (Figure 2a). Additionally, after an hour treatment with 1× MEM AAs, we did not observe any changes in baseline Ca^2+^ levels as measured by ratiometric Ca^2+^ dye Fura-2 (Figure 2b).

Fortunately, we found that the addition of both 1× MEM AA and 1× NEAA’s to HBSS containing 20 mM HEPES decreased the pH to 6.5, and at this pH, we found that both 1× MEM AAs and 1× NEAAs induced an intracellular Ca^2+^ elevation that was completely inhibited by a 1-hour pretreatment with 20 mM of umami antagonist lactisole (Appendix A). The addition of 20 mM lactisole did not alter the pH 6.5 of this solution. As seen in Appendix A, using the resulting pH 6.5 solution, we found that the majority of the Ca^2+^ elevations were from 1× MEM AA’s (~92%) while 1× NEAAs had very little contribution (~6%). While this observation is interesting, it remains to be determined if this is physiologically relevant. We first wanted to study the function of T1R1/3 under normal physiological pH, thus we continued experiments with buffers with pH corrected to 7.4 for the rest of this study.

Though Ca^2+^ signaling was unaffected by 1× MEM amino acids at pH 7.4, we also evaluated cAMP signaling. Utilizing genetic encoded fluorescent biosensor Flamindo2 to measure intracellular cAMP levels [33], we observed that Beas-2Bs responded to the 1× MEM AA mixture with an increase in cAMP production, while 1× NEAAs did not have this effect (Figure 2c). Utilizing a modified Flamindo2 construct with a nuclear localization sequence [33], we also observed that nuclear cAMP levels were also elevated in response to 1× MEM AA treatment (Figure 2d). However, sweet taste receptor agonist sucralose (10 mM) and umami agonist inosine monophosphate (IMP; 10 mM) did not alter intracellular cAMP levels (Figure 2e) though higher concentrations of umami agonist monosodium glutamate (MSG) in line with concentrations used to activate T1R1/3 did increase cAMP (Appendix A). To further examine whether T1R1 or T1R3 were responsible for the cAMP elevations in response to MEM amino acids, we utilized RNAi knockdown.

Using RNAi duplexes, we obtained a ~70% knockdown of *TAS1R3* expression in Beas-2Bs (Figure 2f). Beas-2Bs treated with RNAi showed reduced cAMP elevations relative to cells transfected with non-targeting RNAi in response to 1× MEM AAs (Figure 2g). A knockdown of ~70% *TAS1R3* RNA expression yielded ~70% less cAMP. Since T1R1 dimerizes with T1R3 to comprise the umami taste receptor [9], we also evaluated if knocking down T1R1 had a similar effect on cAMP signaling. We found that a ~75% knockdown of T1R1 (Figure 2h) caused a ~75% reduction in 1× MEM AA generated cAMP production (Figure 2i). Together, these data suggest that both T1R1 and T1R3 are expressed in airway epithelial cells and function to detect at least some of the amino acids in the MEM AA mixture and then regulate cAMP.

Elevations in cAMP alters many downstream signaling pathways, canonically through protein kinase A (PKA) or exchange protein directly activated by cAMP (EPAC). Using Beas-2B cells expressing PKA biosensor AKAR4 [34,35] which acts as a substrate for PKA [36], we did not observe changes in intracellular or nuclear PKA activity with the addition of 1× MEM AA (Figure 2j). Thus, any changes in PKA, if they occur, are below the sensitivity of this biosensor. However, the Flamindo2 biosensor used above is based on the cAMP binding site of EPAC [33] and EPAC activity has been associated with ER Ca^2+^ influx and efflux pathways through modulation of the sarco/endplasmic reticulum Ca^2+^-ATPase (SERCA) and ryanodine (RYR) receptors [37,38,39,40]. Thus, we explored the impact that MEM AA’s and T1R1/3 knockdowns had on ER Ca^2+^ content.

### 3.2. Amino Acids and T1R1/3 Expression Impact ER Ca^2+^ Content

To rigorously test changes in ER Ca^2+^ content, we utilized the SERCA pump inhibitor thapsigargin. Without the constant uptake of Ca^2+^ into the ER, cells treated with thapsigargin will release their ER Ca^2+^ content into the cytosol where it can be measured using the intracellular Ca^2+^ binding dye Fluo-8. The Ca^2+^ will then dissipate into the extracellular space where it will be chelated by EGTA in the surrounding 0-Ca^2+^ HBSS buffer. This allows us to observe the release of total ER Ca^2+^ content without re-uptake of Ca^2+^ across the plasma membrane or ER membranes. The addition of 1× MEM AAs caused a 35% decrease in ER Ca^2+^ content (Figure 3a). As expected, given these findings, the addition of 1× MEM AAs to cultures of Beas-2B’s led to a 50% decrease in Ca^2+^ elevation from 100 µM of histamine treatment (Figure 3b). These results suggest that 1× MEM AAs lowers ER Ca^2+^ levels and impacts downstream Ca^2+^ signaling pathways. We also found that umami agonists IMP and MSG both decreased ER store Ca^2+^ content in Beas-2B cells similarly to MEM AAs (Appendix A).

We have shown that T1R1/3 detects MEM AAs and signals through cAMP in Beas-2Bs. Given the effects of 1× MEM AAs on ER Ca^2+^ modulation, we evaluated the role that T1R1 and T1R3 have on ER Ca^2+^ content using thapsigargin. Surprisingly, we found that knocking down T1R1 (Figure 3c) or T1R3 (Figure 3d) also reduced ER Ca^2+^ content by ~25%. While it is paradoxical that knocking down T1R1/3 expression decreases ER Ca^2+^ content as do the amino acids hypothesized to activate the receptor, it does support that T1R1 and T1R3 are involved in regulation of store Ca^2+^ content. Moreover, when T1R3 was knocked down, MEM AA treatment had no effects on store Ca^2+^ content (Figure 3d). These results demonstrate both a baseline role for T1R1/3 as well as a role for T1R1/3 in detection of amino acids, which ultimately modifies ER Ca^2+^. We further investigated the downstream impact of reduced ER Ca^2+^ content on bitter compound signaling pathways.

Interestingly, in our T1R3 knockdown model, there was an increase in the number of cells that responded to 15 mM denatonium treatment, however each cell released less peak Ca^2+^ compared to the non-targeted RNAi controls. These observations support our findings that T1R3 knockdown cells have reduced ER Ca^2+^ content but also suggest an increase in sensitivity to bitter compound treatment through a related or independent mechanism. Notably, these experiments utilized HBSS without additional amino acids, and thus reflect baseline contributions of T1R3, either via constitutive activity or activation through some intracellular amino acid or metabolite.

While T1R1 and T1R3 have been demonstrated to signal through ERK1/2 and mTORC1 in MIN6 pancreatic b cells [32], we wanted to determine if these pathways were also altering downstream Ca^2+^ signaling pathways. To address this question, Beas-2Bs were pretreated with 10 µM of mTOR inhibitor rapamycin for 1 h, however they did not reveal any changes in Ca^2+^ release with T2R agonist denatonium (Appendix A). These data suggest that the ability of T1Rs to alter ER Ca^2+^ content may be independent of mTORC1, making this a unique pathway in basal airway epithelial cells.

We wanted to determine if cAMP was signaling the decrease in ER Ca^2+^ levels. To approach this question, we tested the impact of a 1-hour pretreatment of Beas-2Bs with either 100 µM of isoproterenol or 20 µM of forskolin. Interestingly, only treatment with isoproterenol decreased ER Ca^2+^ levels (Figure 4a). To verify these findings, we pre-treated Beas-2Bs with 100 µM of isoproterenol for 1 h then utilized 15 mM denatonium to induce Ca^2+^ responses. A population analysis initially suggested there was no difference in Ca^2+^ signaling, as measured by the average peak response of the cultures (Figure 4b). However, individual traces revealed that, in cultures treated with isoproterenol, a greater number of cells responded to the treatment but displayed a lower average peak (Figure 4c). Isoproterenol pretreatment led to a 30% increase in the number of responsive cells, but a 25% decrease in the peak fluorescence per cell (Figure 4d). Thus, while more cells were sensitized to the T2R agonist, those that responded demonstrated a reduced Ca^2+^ elevation. Isoproterenol activates β-adrenergic receptors, which may have other downstream effects, so we opted to also evaluate adenylyl cyclase-activator forskolin as a receptor-independent elevator of cAMP. Once more, there was not a significant difference in the average culture Ca^2+^ response to denatonium in cells treated for 1 h with 20 µM forskolin relative to untreated cells (Figure 4e). However, in these experiments, forskolin treatment elevated the number of cells responsive to denatonium by 30% but did not alter the peak fluorescence per cell (Figure 4f), which supports findings in Figure 4a that forskolin does not alter ER Ca^2+^ content. Together, these data demonstrate that cAMP elevation impacted the sensitivity of cells to denatonium-treatment, but ER Ca^2+^ content reduction may be signaled via other pathways or by localized cAMP microdomains that can be targeted by isoproterenol but not forskolin.

### 3.3. Reduction of ER Ca^2+^ Content via Amino Acid Treatment Hinders Denatonium-Induced Apoptosis

Given the reduction in ER Ca^2+^ levels from 1× MEM AA treatment, we hypothesized that 1× MEM AA pre-treatment would lead to a decrease in T2R Ca^2+^ signaling. To test this, Beas-2Bs were pre-treated with 1× MEM AAs for 1 h (conditions shown above to create reduced ER Ca^2+^ stores) then treated with 15 mM denatonium. To our surprise, 1× MEM AA treatment caused a 30% increase in denatonium-induced Ca^2+^ release based on the population average (Figure 5a). However, with 1× MEM AA treatment, we found that four times the number of cells responded to the denatonium treatment, but each responsive cell elevated Ca^2+^ by only approximately half compared with cells without MEM AA’s (Figure 5b). This supports that each cell has less Ca^2+^ to release, even though more cells are responding to denatonium and releasing Ca^2+^. In the absence of extracellular Ca^2+^, 1× MEM AA pre-treatment (1 h) also reduced denatonium-stimulated Ca^2+^ release by ~50% per responsive cell, demonstrating that MEM AA treatment modifies intracellular Ca^2+^ release rather than Ca^2+^ influx (Figure 5c).

Previously, we have found that denatonium-induced nuclear/mitochondrial Ca^2+^ signaling initiates mitochondrial membrane depolarization and induces apoptosis in basal airway epithelial cells [27]. The concentrations of denatonium necessary to activate nuclear Ca^2+^ showed the same dose-dependency as those necessary to activate apoptosis, ranging from ≥5 to 25 mM of denatonium, with 10 mM being an intermediate dosage [27]. Additionally, blocking this Ca^2+^ signaling utilizing BAPTA-AM prevented the activation of executioner caspases 3/7 [27,41]. Given our observations that 1× MEM AA treatment led to a decrease in the peak Ca^2+^ release per cell, we wanted to determine if this had any impact on the downstream Ca^2+^-signaled apoptosis.

To measure caspase 3/7 activity, we utilized CellEvent. CellEvent, a caspase 3/7 detection reagent, contains a DEVD peptide sequence, targeted by caspase 3 or 7, which when cleaved releases a dye that binds to DNA and fluoresces. After a 1-hour preincubation with 1× MEM AAs in HBSS, Beas-2Bs were treated with 10 mM denatonium in the presence of CellEvent. Cells treated with denatonium + MEM AAs revealed a significantly lower fluorescence, indicating less caspase activity, relative to cells treated with denatonium only (Figure 5d). From fluorescence intensity alone as read on a plate reader, fluorescence in cells treated with denatonium + MEM AAs was not significantly different than cells treated with amino acids only, suggesting a complete block of apoptosis. However, using more sensitive microscopy, the addition of denatonium with amino acids did indeed initiate some caspase 3/7 activity compared with control (amino acids only), but at a far lesser extent than denatonium treatment without amino acids (Figure 5e). These data support our findings that MEM AAs increase the number of cells responsive to denatonium treatment but reduce the absolute Ca^2+^ elevations per cell, and this likely reduces the denatonium-signaled apoptosis.

We further tested the effects of MEM AAs on primary nasal and bronchial basal cells cultured in submersion. Stimulation with MEM AAs but not MEM NEAAs elicited cAMP increases in both primary nasal (Figure 6a) and bronchial (Figure 6b) cells, as measured by Flamindo2. Summary data are shown in Figure 6c. Similarly to the results above, in Beas2B cells we observed that pretreatment with MEM AAs or 10 mM MSG-reduced ER store Ca^2+^ content in primary nasal basal cells, as revealed by thapsigargin stimulation in 0-Ca^2+^ in nasal (Figure 6d) and bronchial (Figure 6e) cells. Summary data are shown in Figure 6f. The presence of amino acids was also associated with reduced denatonium-induced apoptosis (Figure 6g) measured by CellEvent assay, also mirroring results from Beas2B cells. These data confirm the data found above in Beas2B cells and suggest Beas2B cells may be an appropriate model to continue to study T1R1/3 signaling in airway basal cells.

## 4. Discussion

Here, we show that and T1R1 and T1R3 are expressed in basal airway epithelial cells. There, they likely function to detect components of MEM AAs and signal via cAMP elevation. While knocking down T1R1 or T1R3 leads to a reduction in ER Ca^2+^, given the absence of extracellular amino acids in these experiments, these results suggest that T1R1/3 may be constitutively active or may have a role in detecting baseline intracellular amino acid levels. Interestingly, and contrary to initial expectations, treatment of wild-type Beas-2Bs with 1× MEM AAs, suggested by other studies to activate T1R1/3, had a similar outcome as knocking down T1R1/3; 1× MEM AA treatment led to a decrease in ER Ca^2+^ content that nonetheless required T1Rs. We observed that both treating wild-type Beas-2Bs with 1× MEM AAs or knocking down T1R1/3 had similar effects: (1) ER Ca^2+^ content decreased, (2) cultures released less calcium per cell in response to denatonium, (3) and the number of cells responding to denatonium increased. Together, from the perspective of ER Ca^2+^ content, these data suggest that MEM AAs might be disrupting the T1R1/3 activity. If T1R1/3 is providing a baseline regulation of ER Ca^2+^ content, one explanation is that something in the MEM amino acids may be acting as an inverse agonist. Additional research would be necessary to determine that this decrease in ER Ca^2+^ content is solely due to amino acid-T1R1/3 interaction and not the action of another receptor. Furthermore, additional, more detailed pharmacological studies, beyond the scope of this study here, would be needed to test inverse agonism of the MEM-amino acid mixture.

While isoproterenol treatment also led to a decrease in ER Ca^2+^ content, forskolin treatment did not. Given these observations, it seems unlikely that cAMP alone is responsible for the reduction of ER Ca^2+^. However, either knocking down T1R1/3, treating with 1× MEM AA, isoproterenol, or forskolin all led to an increased sensitization of cells to agonists that signal via Ca^2+^ elevations. Thus, we hypothesize that while cAMP elevations may increase the number of cells that respond to agonists that signal via intracellular Ca^2+^ pathways, the reduction of ER Ca^2+^ content may occur through a separate, distinct pathway. Moreover, as there are many transcription factors that are sensitive to cAMP levels, the data showing changes in nuclear cAMP may also suggest a possible role for T1R1/3 in amino acid-induced transcriptional changes, which could be investigated in future studies.

T1R3 has been previously characterized to detect some L-amino acids via heterologous expression studies [9,42]. One study observed that human embryonic kidney cells (HEK’s) expressing murine T1R1+T1R3 were able to detect 50 mM each of the following L-amino acids: Cys, Ala, Gln, Ser, Met, Asn, Gly, Thr, Arg, Val, Leu, His to activate intracellular Ca^2+^ signaling [9]. A separate, but similar study revealed that HEK’s expressing heterologous human T1R1/T1R3 detected 50 mM of each of the following L-amino acids: Ala, Ser, Gln, Asn, Arg, His, Asp, and Glu, while those expressing murine T1R1/3 had notable differences in the detectable amino acid profile [42]. Our 1× MEM AA mixture contained 0.6 mM Arg, 0.1 mM Cys, 0.2 mM His, 0.4 mM iso, 0.4 mM Leu, 0.1 mM Met, 0.2 mM Phe, 0.4 mM Thr, 0.05 mM Trp, 0.2 mM Tyr, and 0.4 mM Val. These concentrations are far lower than what was observed using many overexpression models, but synergistic effects might occur in our study using a combination of amino acids. Our non-essential amino acid mixture contained 0.1 mM of each non-essential L-amino acid. Neither of our mixtures contained glutamine. While T1R1/3 may detect Glu, Ala, Asn, and Ser at higher concentrations than used here, the physiological relevance of such high levels, while important for taste on the tongue, remains unclear in terms basal airway epithelial cells. Future experiments utilizing knockdown models of either T1R1, T1R3, and/or both will be necessary to determine which specific amino acid(s) in our 1× MEM amino acid mixture is/are being detected and if higher dosages of non-essential amino acids are ligands for T1R1/3 in airway cells. The likely synergistic effects of several lower-level amino acids we are observing here suggests this requires highly detailed pharmacology studies using heterologous expression of T1R1/3 that are beyond the scope of this current study.

The physiological agonist (s) activating T1R1/3 in basal airway cells also needs to be determined in order to identify whether this is an immune sensor (e.g., detecting amino acids secreted by pathogens) or a sensor of host metabolic or nutritional state. An important caveat to our study is that we do not yet know the identity of the endogenous agonists. Future work is needed. However, MEM AAs have been used as a T1R1/3 agonist in other studies’ cells [31,32], and the use of receptor knockdown techniques shows the response here is dependent on both T1R1 and T1R3. Thus, the main conclusion that can be drawn here is that T1R1 and T1R3 play a role in the responses studied here. We are not yet able to make conclusions regarding the physiological agonists or context (host defense or nutrient sensing). Nonetheless, this paper is, to our knowledge, the first functional study of T1R1/3 in airway epithelial cells.

We did not observe an increase in intracellular Ca^2+^ release in response to 1× MEM AA treatment at physiological pH. However, when added to a standard HBSS formulation containing 20 mM HEPES without any pH corrections, 1× MEM AA and 1× NEAA reduced the pH from 7.5 to 6.5. We observed Ca^2+^ elevations in response to amino acids when the HBSS was at pH 6.5. At this lowered pH, the majority of the Ca^2+^ response was due to the 1× MEM AAs. This Ca^2+^ elevation was also inhibited by pretreatment with 20 mM T1R3 antagonist lactisole, which did not alter the final pH of the solution. While this suggests a function of T1R1/3 during conditions where pH is lowered, the physiological relevance is not yet determined. Additionally, while we have not observed Ca^2+^ elevations in response to amino acids at a physiological pH; this could also be due to variations in G protein subunits bound to T1R1/3 in our cell models used here. The promiscuity of GPCRs for G protein subunits has been well documented [43,44,45,46]. Future studies of the individual components of the MEM AAs will reveal whether the effects of pH are due to modifications of the T1R1/3 receptor or modifications of the amino acids activating it (e.g., differences in the protonation state of the histidine side chain, which has a pKa near 6.0).

Through detecting amino acids, T1R1/3 was shown to activate ERK1/2 and mTORC1 through a Ca^2+^ signaling pathway in pancreatic b cells [31,32] and in mouse neutrophils [47]. We show that T1R1/3′s ability to alter Ca^2+^ signaling pathways reported here is likely independent of mTOR [31,32] as rapamycin treatment did not alter denatonium and Ca^2+^ signaling like MEM-AAs. While ERK1/2 and mTORC1 may still be activated in our models, this is a distinct, non-canonical pathway of umami receptor signaling. Parallel effects of ERK1/2 and/or mTORC1 activation will be investigated in future studies.

Ultimately, this reduction in ER Ca^2+^ content led to a reduction in intracellular Ca^2+^ elevations signaled by denatonium. Thus, amino acids may play a protective role in reducing apoptosis via the umami T1R1/3 receptor. When we previously measured NO production from bitter compounds such as denatonium, flufenamic acid [27], or diphenhydramine [5], we utilized HBSS containing 1× MEM AAs as a source of arginine for NO production. Given that bitter compounds are still able to elicit a Ca^2+^-signaled NO response even in the presence of 1× MEM AAs, our current work suggests that the addition of 1× MEM AAs to a topical T2R agonist therapeutic may help to diminish apoptotic signaling pathways without preventing stimulation of NO production. Thus, while we do not know the identity of the physiological agonist (s) activating T1R1/3 in airway basal cells, our data do reveal the potential therapeutic utility of 1× MEM AAs activation of T1R1/3 to protect airway epithelial cells against pro-apoptotic insults such as bitter agonists.

## 5. Conclusions

Our findings demonstrate a previously undescribed function for T1R1/3 in airway epithelial cells as a receptor for amino acids that signal through cAMP and as a regulator of ER Ca^2+^ content. Furthermore, the addition of 1× MEM AAs reduced ER Ca^2+^ content and greatly impaired apoptosis signaling via denatonium treatment. Therefore, amino acids or other umami receptor agonists may be useful to employ in therapeutics containing bitter compounds to repress unwanted induction of apoptosis.

## Figures and Tables

**Figure 1 nutrients-15-00493-f001:**
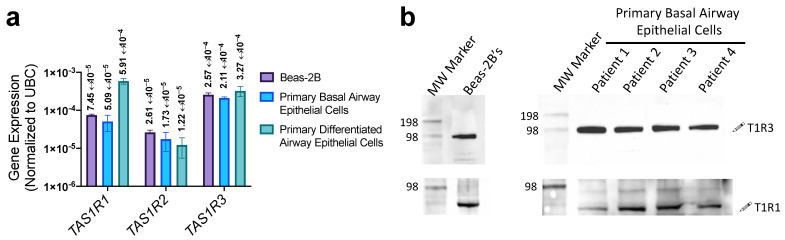
T1R expression in airway epithelial cells. (**a**) qPCR revealed TAS1R1, TAS1R2, and TAS1R3 transcript expression (relative to housekeeping gene UBC) in Beas-2B, basal airway epithelial cells, and differentiated airway epithelial cells. (**b**) Both T1R1 and T1R3 proteins (predicted molecular weight of ~93 kDa) were detected via Western blot using 60 µg protein/lane of Beas-2B or primary basal epithelial cell extract. The visualized protein band representing T1R1 expression migrated slightly lower than predicted 93 kDa but was conserved across cell lines tested. This may be due to incomplete denaturation of the samples prior to SDS-PAGE; freezing or boiling samples for downstream analysis of GPCR’s often leads to formations of aggregates, therefore these samples were processed on the same day without overnight storage or complete denaturation. Bar graphs display mean ± SEM from three experiments.

**Figure 2 nutrients-15-00493-f002:**
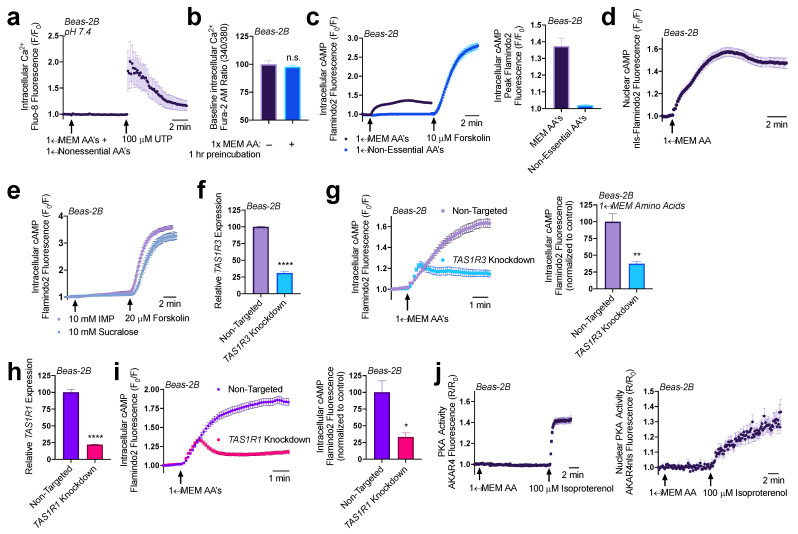
Amino acids and lactisole activate cAMP. (**a**) A pH-corrected mixture of MEM amino acids (AAs) and nonessential amino acids does not induce intracellular Ca^2+^ release in Beas-2Bs. (**b**) 1× MEM amino acids do not alter baseline intracellular Ca^2+^ levels as detected via ratiometric Ca^2+^ dye Fura-2. (**c**) Beas-2Bs expressing cAMP biosensor Flamindo2 reveal that 1× MEM amino acids but not 1× non-essential amino acids alter intracellular cAMP levels. (**d**) 1× MEM AAs also elevate nuclear cAMP levels as measured in Beas-2Bs expressing the Flamindo2 biosensor containing a nuclear localization (nls) sequence. (**e**) Neither T1R2/3 agonist sucralose nor T1R1/3 agonist inosine monophosphate (IMP) alter intracellular cAMP. (**f**) RNAi duplexes targeting TAS1R3 transcript reduce expression by 70% relative to non-targeting scramble controls. (**g**) Knocking down T1R3 impaired detection of MEM amino acids by 70% relative to non-targeting controls. (**h**) RNAi duplexes targeting TAS1R1 transcript reduce expression by 75% relative to non-targeting controls. (**i**) Knocking down T1R1 leads to a 75% reduction in cAMP signaling in response to MEM amino acids relative to non-targeting controls. (**j**) 1× MEM AAs do not increase either intracellular or nuclear PKA activity within the sensitivity range of the AKAR4 biosensor. Traces are representative of ≥3 experiments. Bar graphs show mean ± SEM from ≥3 experiments. Significance determined by *t*-test * *p* < 0.05, ** *p* < 0.01, **** *p* < 0.0001, “n.s.” represents no significance.

**Figure 3 nutrients-15-00493-f003:**
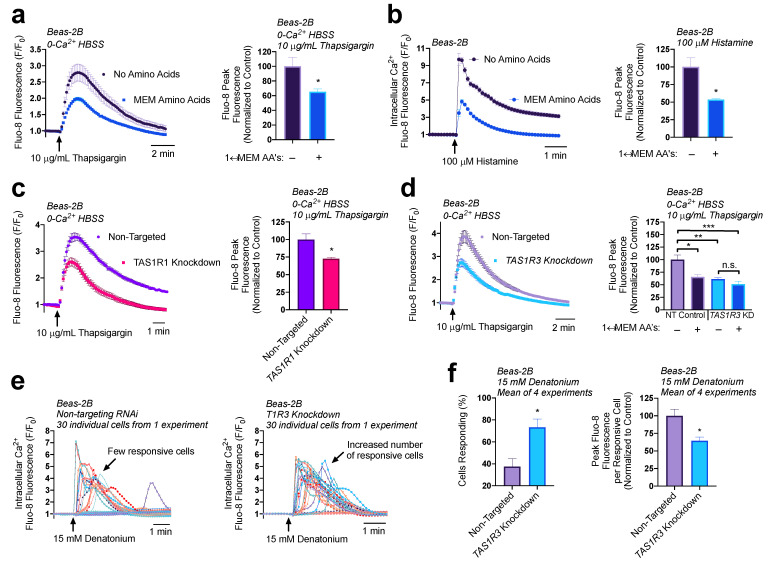
Both T1R1/3 and amino acids impact ER Ca^2+^ content and release. (**a**) Beas-2Bs treated with SERCA-pump inhibitor thapsigargin contain ~40% reduced ER Ca^2+^ content when in the presence of 1× MEM amino acids relative to untreated controls. (**b**) Addition of 1× MEM AA to Beas-2Bs decreased peak Ca^2+^ release via 100 µM histamine treatment by 50%. (**c**) Beas-2Bs with knocked-down T1R1 contained 25% reduced ER Ca^2+^ content relative to non-targeting RNAi controls. (**d**) T1R3 knockdown cultures had 30% reduced ER Ca^2+^ content relative to non-targeting controls, an effect that was not further altered by amino acid treatment. (**e**) Knocking down T1R3 in Beas-2Bs led to an increase in the number of cells responsive to denatonium. (**f**) Quantifying the results from (**e**), knocking down T1R3 caused a doubling of the number of cells responsive to denatonium, while causing a 35% decrease in Ca^2+^ release per responsive cell. Traces are representative of ≥3 experiments. Bar graphs containing only two comparison groups were analyzed via *t*-test; bar graphs containing ≥ 3 groups were analyzed via ANOVA using Bonferroni’s posttest; * *p* < 0.05, ** *p* < 0.01, *** *p* < 0.001, “n.s.” represents no significance.

**Figure 4 nutrients-15-00493-f004:**
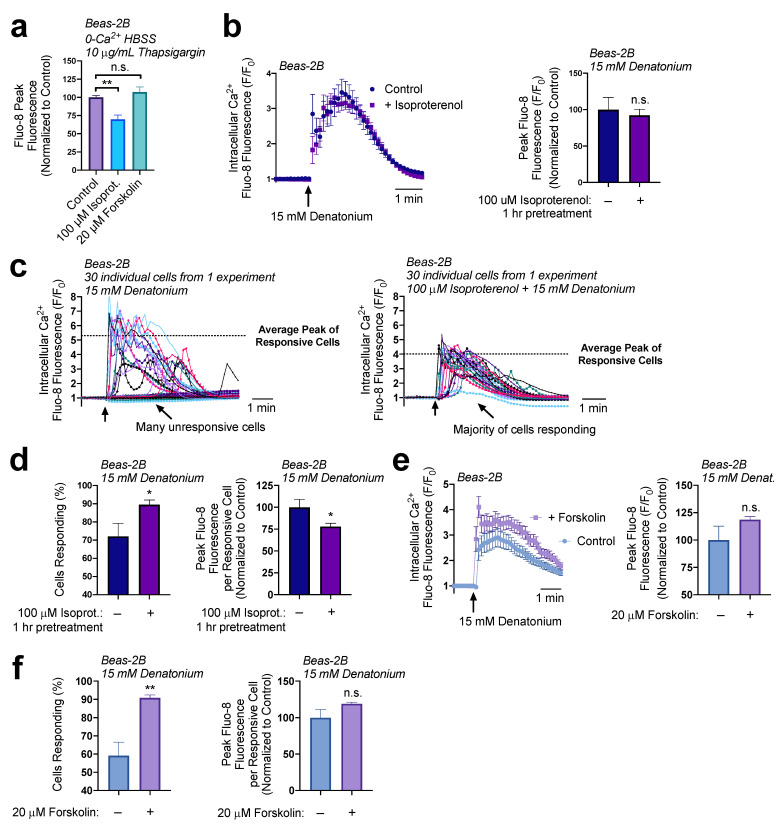
cAMP increases cell responsiveness to denatonium. (**a**) A 1 h preincubation with isoproterenol (100 µM) decreased ER calcium by 30% while forskolin treatment (20 µM) had no significant effect on total ER Ca^2+^. (**b**) A 1 h preincubation with isoproterenol (100 µM) had no effect on the average denatonium-induced Ca^2+^ response relative to untreated samples. (**c**) A closer look at individual cell responses from a representative experiment from (**b**) revealed differences in the number of responsive cells and in the average peak Ca^2+^ of the responsive cells. (**d**) Re-analysis of data from (**b**,**c**) showed that isoproterenol pretreatment (1 h, 100 µM), increased the number of cells responsive to denatonium (15 mM) but decreased peak Ca^2+^ release per cell by 25%. (**e**) Experiments were carried out as in (**b**) but with adenylyl cyclase activator forskolin instead of isoproterenol, with no significant effect on the peak population Ca^2+^ response. (**f**) Like isoproterenol, forskolin increased the percentage of cells responding to denatonium (left graph) but did not significantly affect the peak response per cell (right graph). All traces are representative of ≥3 experiments. Bar graphs show mean ± SEM from ≥3 experiments. Bar graphs with only 2 data points were analyzed by *t*-test, bar graphs of >2 data points analyzed by ANOVA using Tukey’s post-test for multiple comparisons: * *p* < 0.05, ** *p* < 0.01, “n.s.” represents no significance.

**Figure 5 nutrients-15-00493-f005:**
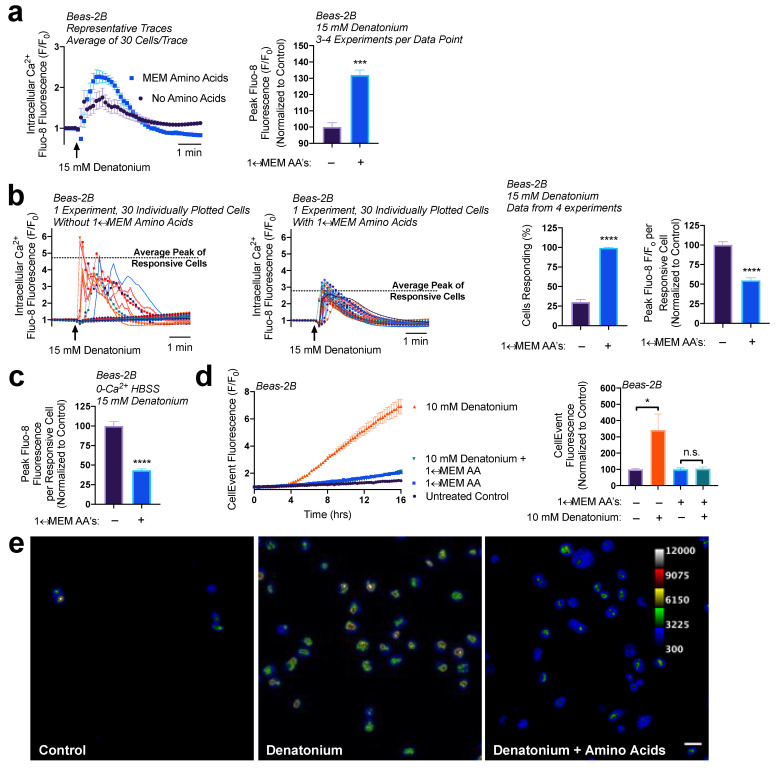
MEM amino acids reduce downstream denatonium-signaled caspase activity. (**a**) Pretreatment of Beas-2Bs with MEM amino acids caused a ~30% increase in denatonium-induced Ca^2+^ elevations. (**b**) Closer analysis of data from (**a**) revealed that amino acid pretreatment increased the number of cells responsive to denatonium, but each cell had ~50% reduced peak Ca^2+^ elevations relative to denatonium-treated cells in HBSS without amino acids. (**c**) Denatonium-induced Ca^2+^ elevations originated from intracellular Ca^2+^ stores in both the presence and absence of MEM amino acids. (**d**) 1× MEM amino acid treatment (1 h preincubation) blocked denatonium-signaled caspase activation as measured on a plate reader. (**e**) Cultures from experiments as in (**d**) visualized by microscopy revealed that some denatonium signaled caspase activity was still present, but significantly reduced. Traces are representative of ≥3 experiments. Bar graphs with only two comparison groups analyzed via *t*-test while bar graphs with >2 data comparison groups analyzed by ANOVA using Sidak’s post-test for paired comparisons: * *p* < 0.05, *** *p* < 0.001, **** *p* < 0.0001, “n.s.” represents no significance.

**Figure 6 nutrients-15-00493-f006:**
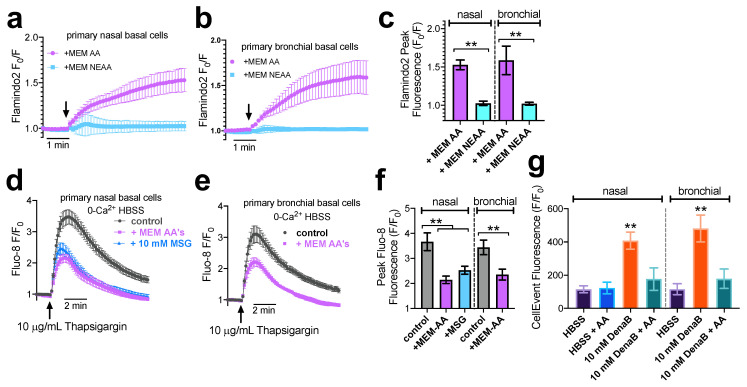
MEM AAs induce both cAMP increases and ER Ca^2+^ store decreases as well as protect against bitter agonist-induced apoptosis in primary nasal and bronchial basal cells. (**a**,**b**) Representative traces (n = 10 cells) showing increase in Flamindo2 F_o_/F (indicating cAMP increases) with MEM AA but not MEM NEAA in primary nasal (**a**) or bronchial (**b**) cells. Experiments were performed as in Figure 2 but with primary cells as indicated. (**c**) Bar graph of results from individual experiments as shown in (**a**) and (**b**). Data are from n = 3 experiments using cells from 3 different patients. (**d**,**e**) Representative traces of thapsigargin-induced Ca^2+^ release as in Figure 3 but with primary nasal (**d**) or bronchial (**b**) basal cells after pre-treatment with MEM AAs or MSG as indicated. (**f**) Bar graph showing summary data from n = 4 experiments as in d and e using nasal cells from n = 4 individual patients and bronchial cells from n = 4 individual patients. (**g**) bronchial cells. (**g**) Bar graph showing CellEvent fluorescence after 6 h with primary nasal (left) or bronchial (right) cells in the presence of denatonium benzoate (DenaB) ± MEM AAs (AA) as indicated. All bar graphs were analyzed by one-way ANOVA with Bonferroni posttest and asterisks show significance compared with respective control; ** *p*< 0.01.

## Data Availability

All data are available by contacting rjl@pennmedicine.upenn.edu.

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
