# Peer review of "Savory Signaling: T1R Umami Receptor Modulates Endoplasmic Reticulum Calcium Store Content and Release Dynamics in Airway Epithelial Cells"

_nutrients, 2023, doi:10.3390/nu15030493_

Round 1

Reviewer 1 Report

In this manuscript, the authors investigated the effects of essential and non-essential amino acids to the cellular responses of airway basal epithelial cells, mainly Beas-2B. While they could not observe the Calcium response, they observed the cAMP elevation by a mixture of essential amino acids (MEM AA), but not by nonessential amino acid mixtures (NEAA), sucralose, and IMP. Because knockdown of TAS1R1 or TAS1R3 reduced the effects of MEM AA, they suggested the MEM effect is mediated by T1R1/3. They further investigated the downstream processes including the effect of bitter compound, denatonium, responses of each cells, ER Calcium contents, etc.

The effect is interesting and give an information on not only Calcium responses but cAMP responses. However, the use of MEM AA is too rough to examine the molecular mechanisms. Because concentrations of AAs in MEM AA and NEAA are different, it is necessary to investigate the effects to each amino acids. Especially, although L-Asp and L-Glu are most effective to the Calcium responses of T1R1/3, it is not reasonable NEAA effects are not investigated so far. It is better to use IMP with L-Asp or LGlu. The arrangements of the figure is necessary to be improved, because the captions are too small to read. They also investigated the effects of pH on the effects of MEMAA and NEAA. If they did the experiments to each amino acids, the effects are more clearly analysed; for example, the protonation states of histidine in MEM AA.

Reviewer 2 Report

1. My question is that what is the relative abundance of T2Rs as compared to umami taste receptors in the nasal airway epithelial cells?

2. How the authors characterized these cells are primary basal epithelial? They must use some markers to confirm these cells.

3. Authors used only BEAS2B cells figure 2 onwards, some result should be repeated in primary basal epithelial cells upon treatments with amino acids and blockers like calcium and cAMP assay.

4. Under pathological conditions like CRS and other airway diseases, which type of amino acids pathogens secreted in majority and what are their concentrations under pathological conditions. My other questions to authors is that the concentrations of amino used in the study is very high, please justify it. 

Round 2

Reviewer 1 Report

The manuscript was much improved; however, the authors claimed more date is necessary to improve completely. I agree to the authors in the point. It is necessary to clarify that cAMP signaling efficiency in human umami-receptor T1R1/3 is different from normal Calcium signaling or not. If the authors showed the results of the same concentrations (0.1mM) of one of the EAA and NEAA (L-Glu or L-Asp), we can compare the downstream mechanisms of the second messengers. If the authors find the downstream mechanisms in the taste signaling and other cells, it will provide new concept for this study field.